# Inhibitory Effective Perturbations of Cilobradine (DK-AH269), A Blocker of HCN Channels, on the Amplitude and Gating of Both Hyperpolarization-Activated Cation and Delayed-Rectifier Potassium Currents

**DOI:** 10.3390/ijms21072416

**Published:** 2020-03-31

**Authors:** Te-Ling Lu, Te-Jung Lu, Sheng-Nan Wu

**Affiliations:** 1School of Pharmacy, China Medical University, Taichung 40402, Taiwan; lutl@mail.cmu.edu.tw; 2Department of Medical Laboratory Science and Biotechnology, Chung Hwa University of Medical Technology, Tainan 71703, Taiwan; lutejung@yahoo.com.tw; 3Institute of Basic Medical Sciences, National Cheng Kung University Medical College, Tainan 70101, Taiwan; 4Department of Physiology, National Cheng Kung University Medical College, Tainan 70101, Taiwan; 5Department of Medical Research, China Medical University Hospital, China Medical University, Taichung 40402, Taiwan

**Keywords:** cilobradine, hyperpolarization-activated cation current, delayed-rectifier K^+^ current, impedance magnitude analysis, current kinetics, pituitary cell, heart cell

## Abstract

Cilobradine (CIL, DK-AH269), an inhibitor of hyperpolarization-activated cation current (*I*_h_), has been observed to possess pro-arrhythmic properties. Whether and how CIL is capable of perturbing different types of membrane ionic currents existing in electrically excitable cells, however, is incompletely understood. In this study, we intended to examine possible modifications by it or other structurally similar compounds of ionic currents in pituitary tumor (GH_3_) cells and in heart-derived H9c2 cells. The standard whole-cell voltage-clamp technique was performed to examine the effect of CIL on ionic currents. GH_3_-cell exposure to CIL suppressed the density of hyperpolarization-evoked *I*_h_ in a concentration-dependent manner with an effective IC_50_ of 3.38 μM. Apart from its increase in the activation time constant of *I*_h_ during long-lasting hyperpolarization, the presence of CIL (3 μM) distinctly shifted the steady-state activation curve of *I*_h_ triggered by a 2-s conditioning pulse to a hyperpolarizing direction by 10 mV. As the impedance-frequency relation of *I*_h_ was studied, its presence raised the impedance magnitude at the resonance frequency induced by chirp voltage. CIL also suppressed delayed-rectifier K^+^ current (*I*_K(DR)_) followed by the accelerated inactivation time course of this current, with effective IC_50_ (measured at late *I*_K(DR)_) or *K*_D_ value of 3.54 or 3.77 μM, respectively. As the CIL concentration increased 1 to 3 μM, the inactivation curve of *I*_K(DR)_ elicited by 1- or 10-s conditioning pulses was shifted to a hyperpolarizing potential by approximately 10 mV, and the recovery of *I*_K(DR)_ inactivation during its presence was prolonged. The peak Na^+^ current (*I*_Na_) during brief depolarization was resistant to being sensitive to the presence of CIL, yet to be either decreased by subsequent addition of A-803467 or enhanced by that of tefluthrin. In cardiac H9c2 cells, unlike the CIL effect, the addition of either ivabradine or zatebradine mildly led to a lowering in *I*_K(DR)_ amplitude with no conceivable change in the inactivation time course of the current. Taken together, the compound like CIL, which was tailored to block hyperpolarization-activated cation (HCN) channels effectively, was also capable of altering the amplitude and gating of *I*_K(DR)_, thereby influencing the functional activities of electrically excitable cells, such as GH_3_ cells.

## 1. Introduction

Cilobradine (CIL, DK-AH269) has been previously demonstrated to suppress hyperpolarization-activated cyclic nucleotide-gated (HCN) channels with an IC_50_ (The concentration of CIL for half-maximal inhibition) of 0.62 μM in mouse sinoatrial node cells (Stieber et al., 2006). CIL could slow the heart rate by decreasing the spontaneous firing rate of the sinoatrial node in the heart (Bois et al., 2011). In telemetric ECG recordings in mice, this compound has also been able to reduce heart rate in a dose-dependent fashion with effective half-maximal dose for the decrease of heart rate (ED_50)_ of 1.2 mg/kg [1,2]. However, it has been notably reported that CIL at the concentrations larger than 5 mg/kg could exert unwanted pro-arrhythmic properties [2]. There is also evidence to reveal that the presence of CIL possesses a positive inotropic action in the mammalian heart [3]. Alternatively, previous observations have revealed its effectiveness in modifying hyperpolarization-elicited hyperpolarization-activated cation current (*I*_h_) in pancreatic α- or β-cells, thereby influencing hormonal secretion [4,5,6]. However, to what extent this compound is capable of resulting in any perturbations on various other types of ionic currents in different types of endocrine cells, such as pituitary cells, has not been thoroughly elaborated.

The magnitude of hyperpolarization-activated cation current (*I*_h_ or funny current [*I*_f_]) has been growingly considered to be the prominent determinant of repetitive electrical activity inherently in heart cells and in an array of neurons and neuroendocrine or endocrine cells [5,7,8,9,10,11,12,13,14,15,16]. This current is characterized by a mixed inward Na^+^/K^+^ one with a slowly activating property during long-lasting membrane hyperpolarization, and it is subject to being blocked by various compounds, such as CsCl, ivabradine, and zatebradine [9,14,15,16,17,18,19].

As described in a number of previous studies [7,9,10,12,13,14], the increased amplitude of *I*_h_ can act to depolarize membrane potential to the threshold required for elicitation of the action potential in electrically excitable cells. This type of ionic current has been regarded to be carried by channels of the hyperpolarization-activated cyclic nucleotide-gated (HCN1-4) gene family, which belongs to the superfamily of voltage-gated K^+^ channels and cyclic nucleotide-gated channels [8,12,18,20]. 

Voltage-gated K^+^ (K_V_) channels play essential roles in determining membrane excitability, and the delayed rectifier K^+^ channels, such as K_V_3 (KCNC) and K_V_2 (KCNB) channels, are ubiquitous in endocrine cells [21,22,23,24]. A causal relationship between the K_V_3 or K_V_2 channel and the delayed rectifier K^+^ current (*I*_K(DR)_) has been previously established [21,23,25,26]. The K_V_ channels from the K_V_3.1-K_V_3.2 types, the biophysical properties of which exhibit to have positively shifted voltage dependency and fast deactivation rate, are the major determinants of *I*_K(DR)_ identified in the pituitary tumor (GH_3_) cells [23,24,26]. However, to the best of our knowledge, whether and how CIL or other structurally similar compounds affect any modifications on these types of K^+^ currents (e.g., *I*_K(DR)_) remains largely unexplored.

Therefore, in light of the above-described considerations, we intended to address the possibility that CIL exerts the perturbing actions on different types of ionic currents, which include hyperpolarization-activated cation current (*I*_h_), delayed-rectifier K^+^ current (*I*_K(DR)_), and voltage-gated Na^+^ current (*I*_Na_) identified in pituitary tumor (GH_3_) cells. The *I*_K(DR)_ in H9c2 cardiac cells was also tested during the exposure to CIL, ivabradine, or zatebradine. The experimental results presented here reflect that, distinguishable from the actions of ivabradine or zatebradine on *I*_K(DR)_, the presence of CIL was capable of suppressing *I*_K(DR)_ effectively in a concentration-, time-, and state-dependent manner, despite the ability of it, ivabradine, or zatebradine to suppress the amplitude of *I*_h_.

## 2. Results

### 2.1. Effects of CIL on Hyperpolarization-Activated Cation Current (I_h_) Recorded from Pituitary Tumor (GH_3_) Cells

In the experiments, we initially tested whether this compound had any perturbations on the amplitude and gating of this current identified in these cells. In order to preclude the contamination of most Ca^2+^-activated K^+^ currents [22,27,28], we immersed the cells in Ca^2+^-free, Tyrode’s solution, and during the measurements, we backfilled the recording pipette with a K^+^-containing solution. As the whole-cell current recordings were attained, we maintained the examined cell at −40 mV, and the 2-s hyperpolarizing voltage-clamp step to various potentials was later applied to elicit the slowly inwardly directed *I*_h_, which displayed an inward-rectifying property. As illustrated in Figure 1A, within 2 min of exposing cells to CIL (3 μM), the *I*_h_ amplitude in response to various hyperpolarizing steps was robustly decreased. In this set of experiments, we applied a two-step voltage protocol in which various conditioning potentials ranging from −30 to −120 mV with 2 sec in 10-mV step preceding the 2-s hyperpolarizing pulse to −120 mV were delivered to the examined cell. The averaged current density versus voltage relation of *I*_h_ measured at the end of each conditioning pulse with or without the addition of CIL (3 μM) is illustrated in Figure 1B. For example, at the level of −120 mV, the presence of CIL (3 μM) decreased *I*_h_ density from 15.9 ± 1.3 to 8.7 ± 1.1 pA/pF (n = 8, *p* < 0.05). After washout of the agent, current density returned to 15.2 ± 1.2 pA/pF (n = 7, *p* < 0.05). Concurrently, during cell exposure to 3 μM CIL, the macroscopic *I*_h_ conductance measured at the potentials ranging between −90 and −120 mV was decreased to 7.57 ± 0.23 nS (n = 7, *p* < 0.05) from a control value of 16.27 ± 0.34 nS (n = 7). The time constant (τ) of *I*_h_ activation at the level of −120 mV was concomitantly lengthened to 1754 ± 117 msec (n = 7, *p* < 0.05) from a control value of 1354 ± 143 msec (n = 7).

The steady-state activation curve of *I*_h_ with or without CIL addition was further analyzed by using this two-step voltage protocol (Figure 1C). In the control—*V*_1/2_ = −94.4 ± 1.8 mV and *q* = 4.9 ± 0.1 *e* (n = 7), while, in the presence of 3 μM CIL—*V*_1/2_ = −104.3 ± 1.9 mV and *q* = 5.1 ± 0.1 *e* (n = 7). The present observations thus reflected that the presence of CIL did not merely reduce maximal *I*_h_ conductance but also distinctly shifted the midpoint of inactivation curve along the voltage axis to a hyperpolarizing potential by approximately 10 mV; however, the gating charge of the activation curve entailed did not differ significantly between the absence and presence of 3 μM CIL. The concentration-dependent relation of the CIL action on *I*_h_ amplitude was derived and is then illustrated in Figure 1D. The IC_50_ value required for its inhibitory effect on the *I*_h_ amplitude measured at the level of −120 mV was estimated to be 3.38 μM.

### 2.2. Effects of CIL on the Impedance Magnitude (IM) as a Function of Frequency (i.e., IM Plot)

We next evaluated the relationship between impedance magnitude and oscillating frequency by applying membrane hyperpolarization at different intensities of 1–5 mV chirp voltage with the decreasing frequencies (1–300 Hz). This could allow an evaluation of the relationship between the *I*_h_ magnitude and oscillatory behavior of the current [29,30]. As shown in Figure 2, as GH_3_ cells were exposed to different CIL concentrations, the density of *I*_h_ was evidently decreased. Upon step hyperpolarization to −120 mV with a linear decrease in the frequency from 300 to 1 Hz, the addition of 3 μM CIL increased the impedance magnitude taken at the level of resonance frequency (f_R_) from 0.063 ± 0.005 to 0.17 ± 0.01 GΩ (n = 8, *p* < 0.05). No discernible change in the f_R_ value (i.e., around 12.2 Hz) was, however, altered during exposure to different CIL concentrations. Consequently, in addition to the inhibition of *I*_h_ density, the impedance amplitude profile of *I*_h_ in response to chirp voltage could be modulated by the presence of CIL.

### 2.3. Averaged Current Density Versus Voltage Relationship of I_K(DR)_ Produced by the Presence of CIL

We further examined whether CIL was able to alter the amplitude and gating of *I*_K(DR)_ in GH_3_ cells. In this set of experiments, cells were bathed in Ca^2+^-free Tyrode’s solution containing 1 μM tetrodotoxin, and the recording pipette was backfilled with a K^+^-containing solution. As depicted in Figure 3, the addition of 3 μM CIL was able to consistently suppress the amplitude of outwardly directed *I*_K(DR)_ elicited by 1-s voltage pulses ranging between −60 to +60 mV. In addition to the decreased current amplitude, the inactivation time course of *I*_K(DR)_ in response to 1-s step depolarization was observed to become noticeably shortened. For example, when CIL (3 μM) was applied to the bath, at the level of +50 mV, current amplitude measured at the beginning (i.e., peak *I*_K(DR)_) of step depolarization decreased from 23 ± 3% to 20.9 ± 1.1 pA/pF from a control value of 27.3 ± 1.7 pA/pF (n = 8, *p* < 0.05) (Figure 3B), while that at the end (i.e., late *I*_K(DR)_) of voltage pulse decreased 44 ± 4% from 23.4 ± 1.5 to 13.0 ± 1.1 pA/pF (n = 8, *p* < 0.05) (Figure 3C). After the washout of the compound, the amplitude of late *I*_K(DR)_ returned to 22.7 ± 1.4 pA/pF (n = 7, *p* < 0.05). Moreover, as cells were exposed to 3 μM CIL, the macroscopic peak *I*_K(DR)_ conductance (i.e., at the beginning of each depolarizing pulse) measured at the voltages ranging between 0 and +60 mV was noticeably diminished by only 27 ± 3% to 8.3 ± 0.1 nS (n = 8, *p* < 0.05) from a control value of 11.3 ± 0.2 nS (n = 8); however, whole current conductance taken at the end of each 1-s depolarizing pulse was lowered by 49 ± 3% from 8.1 ± 0.2 to 4.1 ± 0.1 nS (n = 8, *p* < 0.05).

### 2.4. Kinetic Evaluation of the CIL-Inhibited I_K(DR)_ Observed in GH_3_ Cells

To provide a quantitative estimate of the CIL-mediated inhibition of *I*_K(DR)_, we further analyzed the time constants (τ) for the relative block of *I*_K(DR)_ (i.e., (*I*_control_ – *I*_CIL_)/*I*_control_) taken from these cells. In the first step, we appropriately fitted the time courses for the relative blocking of the currents to a single-exponential function. The concentration dependence of the relative *I*_K(DR)_ blocking elicited by 1-s membrane depolarization is illustrated in Figure 4A–C. It is noticeable from these data that the presence of CIL led to a linear concentration-dependent increase in the rate (1/τ) of relative blocking on *I*_K(DR)_. For example, when cells were depolarized to +50 mV from a holding potential of −50 mV, the τ values of the *I*_K(DR)_ relative blocking achieved in the presence of 0.3 or 1 μM CIL were fitted to a single exponential with the value of 175 ± 13 msec (n = 8) or 145 ± 12 msec (n = 8), respectively. 

On the premise that the first-order minimal kinetic scheme elaborated under Materials and Methods occurred, a linear relationship between 1/τ and the CIL concentration [CIL] with an R^2^ value of 0.96 was taken (Figure 4C). The resultant blocking or unblocking rate constant produced during exposure to different CIL concentrations was estimated to be 1.18 sec^−1^μM^−1^ or 4.45 sec^−1^, respectively. Therefore, we yielded the value of the dissociation constant (*K*_D_ = *k*_−1_/*k*_+1_^*^) with 3.77 μM. By comparison, as depicted in Figure 4D, the IC_50_ value required for the inhibition of *I*_K(DR)_, measured at the beginning (peak *I*_K(DR)_) or end (late *I*_K(DR)_), of depolarizing pulse was 9.32 or 3.54 μM, respectively. The *K*_D_ value required for the CIL-mediated block of *I*_K(DR)_ was very close to the observed IC_50_ measured at the end of depolarizing steps (late *I*_K(DR)_). 

### 2.5. Steady-State Inactivation Curve of I_K(DR)_ Obtained in the Presence of CIL

Alternatively, we further characterized the CIL effectiveness on the regulation of the inactivation curve of *I*_K(DR)_ in GH_3_ cells. The steady-state inactivation curves of *I*_K(DR)_ in the presence of CIL 1 and 3 μM were further constructed and are hence illustrated in Figure 5. In these whole-cell recordings, a 1-s conditioning pulse to various potentials was applied to precede the 1-s test pulse to +50 mV from a holding potential of −50 mV. The relationship between the conditioning potentials and the normalized amplitude of *I*_K(DR)_ during exposure to different CIL concentrations (1 and 3 μM) was derived and constructed with a Boltzmann function. In the control, voltage for half-maximal inactivation (*V*_1/2_) and corresponding gating charge (*q*) were −25.1 ± 2.2 mV and 2.7 ± 0.3 *e*, while in the presence of 1 μM or 3 μM CIL, the values of *V*_1/2_ and *q* were −27.7 ± 2.1 mV and 2.7 ± 0.2 *e* (n = 7), or −35.1 ± 2.5 mV and 2.7 ± 0.2 *e* (n = 7), respectively. Therefore, as the CIL concentration increased from 1 to 3 μM, the midpoint of the inactivation curve of the current was evidently shifted along the voltage axis toward hyperpolarizing voltage by approximately 7 mV. No pronounced difference in the gating charge was, however, demonstrated during exposure to 1 or 3 μM CIL.

### 2.6. Effect of CIL on Recovery of I_K(DR)_ from Inactivation in GH_3_ Cells

The effect of CIL on the recovery of *I*_K(DR)_ from the inactivation was evaluated with a two-step voltage-clamp protocol. In these measurements, a 300-msec conditioning step to +50 mV (i.e., 1st voltage-clamp step) inactivated most of the *I*_K(DR)_, and the recovery of *I*_K(DR)_ from the inactivation at the holding potential of −50 mV was examined at different times with a test step (+50 mV, 300 msec, i.e., 2nd voltage-clamp step), as shown in Figure 6. During cell exposure to 1 μM or 3 μM CIL, the time course of recovery from the current inactivation was fitted to a single exponential function with a τ value of 538 ± 9 msec (n = 8) or 613 ± 11 msec (n = 8), respectively. Our results, therefore, reflected that as the CIL concentration increased from 1 to 3 μM CIL, recovery from the *I*_K(DR)_ inactivation observed in GH_3_ cells significantly became prolonged.

### 2.7. Inhibitory Effect of CIL on the Steady-State Inactivation Curve of I_K(DR)_ Evoked by 10-s Conditioning Pulses

We further characterized the voltage dependence of 10-s long depolarization-elicited *I*_K(DR)_ with or without the application of CIL. The steady-state inactivation curve of *I*_K(DR)_ following the addition of 1 or 3 μM CIL is shown in Figure 7. In the presence of 1 μM CIL, the value of *V*_1/2_ or *q* was −45.1 ± 2.1 mV or 2.6 ± 1.9 *e* (n = 7), while, in the presence of 3 μM CIL, the value of *V*_1/2_ or *q* was −55.1 ± 2.2 mV or 2.5 ± 1.8 *e* (n = 7). Under our experimental conditions, the inactivation curve of this current elicited by 10-s command potentials was shifted to a hyperpolarized potential by 10 mV as the CIL concentration increased from 1 to 3 μM; however, the gating charge of the current did not differ significantly between the presence of 1 and 3 μM CIL. 

### 2.8. Failure of CIL to Suppress Voltage-Gated Na^+^ Current (I_Na_) Evoked by Brief Depolarization

We further tested whether CIL was capable of modifying *I*_Na_ identified in GH_3_ cells. In these experiments, cells were suspended in Ca^2+^-free Tyrode’s solution, and the Cs^+^-containing solution was filled up in the recording pipette. As shown in Figure 8, as cells were exposed to 3 μM CIL, the peak density of *I*_Na_ in response to brief membrane depolarization was not altered. However, in the continued presence of 3 μM CIL, subsequent application of 3 μM A-803467 or 3 μM tefluthrin to the bath medium was able to suppress or enhance the peak *I*_Na_ density, respectively. A-803467 has been reported to be a selective blocker of Na_V_1.8 channels [31], while tefluthrin, a pyrethroid, can activate *I*_Na_ [32]. Similarly, further addition of 1 μM tetrodotoxin, still in the presence of 3 μM CIL, was found to fully decreased peak *I*_Na_ density. In this regard, unlike the *I*_h_ or *I*_K(DR)_, the *I*_Na_ inherently in these cells revealed no responsiveness to CIL.

### 2.9. Effect of CIL, Ivabradine, or Zatebradine on I_K(DR)_ Recorded from Heart-Derived H9c2 Cells

An additional step of experiments was conducted to examine if there is any possible modification by CIL, ivabradine, or zatebradine on depolarization-elicited *I*_K(DR)_ identified in H9c2 cells. As illustrated in Figure 9, the presence of 3 μM CIL was effective at suppressing the *I*_K(DR)_ amplitude and raising the rate of the current inactivation, while that of 3 μM ivabradine or 3 μM zatebradine mildly suppressed *I*_K(DR)_ with no noticeable modifications in the inactivation time course of this current. For example, as the examined cells were depolarized from −50 to +50 mV, CIL (3 μM) decreased late *I*_K(DR)_ by 64.4 ± 2.8% from 1160 ± 32 to 413 ± 19 pA (n = 8, *p* < 0.05), while zatebradine (3 μM) decreased late *I*_K(DR)_ by 32.6 ± 2.1% from 1107 ± 29 to 746 ± 22 pA (n = 8, *p* < 0.05) with minimal change in the inactivation time course of the current. Similar to the results disclosed above in GH_3_ cells, the presence of CIL could significantly but differentially suppress the peak and late amplitude of *I*_K(DR)_ in cardiac H9c2 cells; however, ivabradine or zatebradine slightly suppressed *I*_K(DR)_ amplitude.

## 3. Discussion

In good agreement with the findings of earlier studies [16,33,34], the present observations disclosed that the presence of CIL was able to suppress the amplitude of hyperpolarization-elicited *I*_h_, as well as to slow the rate of the current activation (Figure 1). There was a distinct hyperpolarizing shift in the midpoint of the activation curve of *I*_h_ in its presence with no alterations in the gating charge of the current (Figure 1). In this regard, the magnitude of its depressant action on *I*_h_ density in native cells could be strongly governed by factors, such as the CIL concentrations achieved, the pre-existing resting potential, the firing patterns of excitable cells, or a combination. 

The effective IC_50_ required for the CIL-mediated inhibition of *I*_h_ was calculated to be 3.38 μM (Figure 1D), a value that tends to be higher than those for either its suppression of HCN channels identified in mouse sinoatrial cells (i.e., 0.62 μM) [2] and in cardiac Purkinje fibers [17] or the slowing of heart rate (i.e., 0.023 μM) [35,36,37]. It has been described that the HCN2, HCN3, or mixed HCN2+HCN3 channels are functionally expressed in GH_3_ cells or other types of endocrine cells [9,10,11,12,13,14,15]. However, in GH_3_ cells, we were unable to detect instantaneous current, which was thought to be functionally expressed in the HCN2 channel, as described previously [38]. Nevertheless, the affinity of CIL for the HCN channels identified in GH_3_ cells could, therefore, be less than that for the HCN channels in heart cells. Alternatively, the heart-rate response to CIL has been previously noticed with a significant delay with respect to the time course of CIL in plasma [37]. Therefore, to what extent CIL-mediated inhibition of *I*_K(DR)_ contributes to such delayed slowing in heart rate remains to be further delineated.

According to the impedance magnitude (IM) plot observed here (Figure 2) [29,30], as chirp voltage with the decreasing frequencies was applied to the cells, the resonance frequency (f_R_) with around 12.2 Hz was calculated. Cell exposure to CIL was noted to increase the IM value measured at the level of resonance frequency, although no discernible change in resonance frequency was detected. As described previously [29,30], the observed effects of CIL, therefore, led us to propose that changes in the sub-threshold oscillating current during *I*_h_ elicitation could be raised in the presence of CIL.

The inhibitory action of CIL on *I*_K(DR)_ in GH_3_ and H9c2 cells was observed to correlate over time with a conceivable raise in the inactivation rate of the current in response to step depolarization (Figure 3 and Figure 4). The values of *K*_D_ (calculated from the first-order reaction scheme; Figure 4C) or IC_50_ (needed for the CIL-induced block of late *I*_K(DR)_; Figure 4D) recorded from GH_3_ cells were essentially indistinguishable (i.e., around 3 μM). Moreover, as the CIL concentration increased, the recovery from *I*_K(DR)_ block was prolonged (Figure 6). However, the multiple-binding site scheme for CIL action on *I*_K(DR)_ should not be excluded, although a plausible interpretation for such block was adequately made according to the minimal binding scheme, as elaborated under Materials and Method. Nonetheless, the suppression of *I*_h_ and *I*_K(DR)_ produced by CIL might concertedly affect the electrical behaviors of endocrine or neuroendocrine cells, which are present in vivo. Indeed, previous studies have demonstrated the presence of *I*_h_ and *I*_K(DR)_ in endocrine cells, such as pancreatic α- or β-cells [4,5,6,11,39]. The *I*_K(DR)_ presented herein (i.e., K_V_2.1-encoded current) has been reported to be functionally enriched in different types of heart cells [25,26,40,41]. Consequently, it is possible that the mediation by CIL of *I*_K(DR)_ inhibition influences the heart function to a certain extent, as previously reported with respect to the effectiveness of aconitine, a potent cardiotoxin, in the amplitude and gating of depolarization-triggered *I*_K(DR)_ in heart cells [41].

The results obtained also demonstrated that the shift in the midpoint of the inactivation curves of 1-s (Figure 5) or 10-s (Figure 7) conditioning potential-induced *I*_K(DR)_ in response to CIL was concentration-dependent. In contrast, we failed to detect any changes in the gating charge of the current taken during exposure to different CIL concentrations. However, the response of the concentration of this effect occurred at the concentrations similar to those which suppress *I*_K(DR)_ amplitude and elevate the inactivation time course of the current. In this scenario, distinct from the effects of zatebradine or ivabradine (Figure 9), the inhibitory action of CIL on *I*_K(DR)_ could thus reflect its higher affinity bindings to K_V_-channel states, favored mainly at depolarized potentials, such as the open or inactivated state of the channel, although the detailed gating conformational changes of the channel in the presence of CIL need to be further characterized. 

Alternatively, it is important to emphasize that by comparison, the CIL molecule (https://pubchem.ncbi.nlm.nih.gov/compound/16078969) tends to become less conformationally flexible than ivabradine or zatebradine because a six-membered piperidine ring residing in the molecule is incorporated into the flexible chain. We thus hypothesized that changes in conformation might alter the spatial relationship between the pharmacophoric (i.e., functional) groups in the CIL molecule and thereby have an influence on its additional interaction with the *I*_K(DR)_, although these three compounds are bioisoteric compounds, which are benzazepinone derivatives, and able to produce the inhibitory actions on *I*_h_ that are structurally related to each other. In other words, incorporation of the six-membered piperidine ring into the flexible chain in the zatebradine molecule could be entailed to produce the CIL molecule with strong inhibition of *I*_K(DR)_ along with accelerated current inactivation in GH_3_ and H9c2 cells.

It is also necessary to point out that the potential of HCN blockers in cardiac pathologies and in the management of pain is well established and that the compounds able to increase *I*_h_ (e.g., oxaliplatin) may have the therapeutic potential [42,43]. Caution thus entails being raised in attributing the action of CIL on either cardiac function or neuronal networks widely to the block of *I*_h_ [2,18,19,30,34,35,36,44,45]. Moreover, the exposure to CIL has been reported to modulate balance function, given that it might concertedly influence both functional HCN channels (i.e., HCN1) in vestibular hair cells of the inner ear [44] and the K_v_3.1 channels, which are enriched in the auditory pathway [46].

## 4. Materials and Methods

### 4.1. Chemicals, Drugs, and Solutions

Cilobradine (CIL, DK-AH269, 3-[[(3S)-1-[2-(3,4-dimethoxyphenyl)ethyl]-3-piperidinyl]methyl]-1,3,4,5-tetrahydro-7,8-dimethoxy-2H-3-benzazepin-2-one, monohydrochloride, C_28_H_38_N_2_O_5_·HCl, https://pubchem.ncbi.nlm.nih.gov/compound/16078969) was acquired from Cayman Chemical (Excel Biomedical, Taipei, Taiwan), and tefluthrin (Tef), tetraethylammonium chloride (TEA), tetrodotoxin, and ivabradine (IVA, https://pubchem.ncbi.nlm.nih.gov/compound/Ivabradine) were from Sigma-Aldrich (Merck Ltd., Taipei, Taiwan), while A-803467 (5-(4-chlorophenyl)-N-(3,5-dimethoxyphenyl)furan-2-carboxamide) and zatebradine (ZAT, UL-FS49, C_26_H_36_N_2_O_5_, https://pubchem.ncbi.nlm.nih.gov/compound/Zatebradine) were from Tocris (Union Biomed Inc., Taipei, Taiwan). Unless otherwise noted, culture media, fetal bovine serum, horse serum, L-glutamine, and trypsin/EDTA were obtained from HyClone^TM^ (Thermo Fisher, Level Biotech, Tainan, Taiwan), while other chemicals or reagents were of analytical grade.

The HEPES-buffered normal Tyrode’s solution used in this study had a composition, which comprised (in mM): NaCl 136.5, KCl 5.4, CaCl_2_ 1.8, MgCl_2_ 0.53, glucose 5.5, and HEPES 5.5 titrated to pH 7.4 with NaOH. For the measurement of *I*_h_, *I*_K(DR)_, or *I*_Na_, we bathed cells with Ca^2+^-free Tyrode solution. For studies on *I*_h_ or *I*_K(DR)_, we backfilled with the intracellular solution (in mM): K-aspartate 130, KCl 20, KH_2_PO_4_ 1, MgCl_2_ 1, EGTA 0.1, Na_2_ATP 3, Na_2_GTP 0.1, and HEPES 5 titrated to pH 7.2 with KOH, while to measure *I*_Na_ and to avoid the contamination of K^+^ currents, K^+^ ions in the pipette solution were substituted for Cs^+^ ions. All solutions were prepared using demineralized water from a Milli-Q water purification system (Merck, Ltd., Taipei, Taiwan). The pipette solution and culture medium were often filtered on the day of use with an Acrodisc^®^ syringe filter with 0.2-μm Supor^®^ membrane (Bio-Check; New Taipei City, Taiwan).

### 4.2. Cell Preparations

GH_3_ pituitary tumor cells, acquired from the Bioresource Collection and Research Center ([BCRC-60015, https://catalog.bcrc.firdi.org.tw/BcrcContent?bid=60015]; Hsinchu, Taiwan), were seeded in Ham’s F-12 medium supplemented with 15% horse serum (*v/v*), 2.5% fetal bovine serum (*v/v*), and 2 mM *L*-glutamine. To facilitate their differentiation, GH_3_ cells were transferred to a serum-free, Ca^2+^-free medium. The H9c2 cell line, originally derived from embryonic rat ventricles, was also ordered from the Bioresource Collection and Research Center (BCRC-60096; https://catalog.bcrc.firdi.org.tw/BcrcContent?bid=60096). Cells were grown in Dulbecco’s modified Eagle’s medium supplemented with 10% fetal bovine serum (*v/v*) and 2 mM L-glutamine. GH_3_ or H9c2 cells were plated in 100-mm culture dishes (10^6^ cells/dish) and maintained at 37 °C in a humidified environment of 5% CO_2_/95% air. They were sub-cultured weekly, and fresh media were replenished every 23 days to remove non-adhering cells that were unhealthy. Subcultures were obtained by trypsinization (0.025% trypsin solution [HyClone^TM^], containing 0.01% sodium *N*,*N*-diethyldithiocarbamate and EDTA).

### 4.3. Electrophysiological Measurements

On the day of each experiment, GH_3_ or H9c2 cells were harvested, and an aliquot of cell suspension was later transferred on a custom-built recording chamber mounted on the stage of inverted Leica DM-IL microscope (Pentad, Hsinchu, Taiwan), which was coupled to a video camera system with magnification up to 1500×. Cells were bathed at room temperature (20–25 °C) in normal Tyrode’s solution, the composition of which is detailed above. The pipettes used were fabricated from Kimax-51 borosilicate capillaries of 1.5-mm outer diameter (#34500; Kimble; Dogger, New Taipei City, Taiwan) by using either a P-97 Flaming/Brown programmable puller (Sutter; Upwards Biosystems, Taipei, Taiwan) or a PP-83 vertical puller (Narishige; Taiwan Instrument Co., Taipei, Taiwan). The tip resistance was in the range of 3–5 MΩ when the electrode was backfilled up with different internal solutions detailed above. Ion currents were measured in the whole-cell mode of standard patch-clamp technique by using either an Axopatch 200B (Molecular Devices, Sunnyvale, CA, USA) or an RK-400 (Bio-Logic, Claix, France) patch amplifier [15,27,47]. The recorded area on the vibration-free table was shielded by using a Faraday cage (Scitech, Seoul, South Korea) to minimize mechanical noise. All potentials were corrected for junction potential (-13.1 ± 2 mV, n = 14 GH_3_ cells; −13.2 ± 2 mV, n = 12 H9c2 cells) that would appear when the composition in the bath solution differed from that in the pipette internal solution, and whole-cell experimental results were corrected by this potential. 

### 4.4. Data Recordings

The potential and current signals were monitored on an HM-507 oscilloscope (Hameg, East Meadow, NY, USA) and digitally stored online in an Acer SPIN-5 touchscreen laptop computer (SP513-52N-55WE; Taipei, Taiwan) at 10 kHz through a 12-bit resolution Digidata 1440A interface (Molecular Devices, Sunnyvale, CA, USA). During the measurements, the latter device was controlled by pCLAMP 10.7 software (Molecular Devices) run under Microsoft Windows^TM^ 10 (Redmond, WA, USA). The laptop computer used was put on the top of the adjustable Cookskin stand (Ningbo, Zhejiang, China) for efficient operation during the experiments. Cell-membrane capacitance of 18–39 pF (29.2 ± 4.4 pF, n = 26 GH_3_ cells) or 23–43 pF(31.4 ± 5.4 pF, n = 21 H9c2 cells) was compensated. Series resistance, always in the range of 5–15 Mτ, was electronically compensated to 80–95%. The linear passive leak currents were subtracted using a P/4 regimen. Current signals were low-pass filtered at 2 kHz with an FL-4 four-pole Bessel filter (Dagan, Tainan, Taiwan) before being digitized. Through digital-to-analog conversion, different pCLAMP-generated voltage-clamp profiles, including chip voltage protocol, were specifically designed to examine either the current-voltage (*I-V*) relations of ionic currents (e.g., *I*_h_ or *I*_K(DR)_) or the activation curve of the currents. To evaluate possible modulations of *I*_h_ in the subthreshold oscillating frequency during varying CIL concentrations, the 2-s stimulating protocol of sinusoidal voltage (i.e., chirp stimulation), in which there is a linear increase in the frequency between 1 and 300 Hz, was designed and incorporated to the different levels of step hyperpolarization. After the signals were digitally achieved, we subsequently analyzed them using varying analytical tools that include LabChart 7.0 program (AD Instruments; KYS Technology, Tainan, Taiwan), OriginPro 2016 (Microcal; Sherman, Taipei, Taiwan), or custom-made macro procedures built under Microsoft Excel^®^ 2013 (Redmond, WA, USA). 

### 4.5. Data Analyses

To assess concentration-dependent inhibition of CIL on the amplitude of *I*_h_ or *I*_K(DR)_, we immersed cells in Ca^2+^-free Tyrode’s solution, the composition of which was elaborated above. For the studies on *I*_h_, the examined cell was hyperpolarized from a holding potential of −40 to −120 mV with a duration of 1 sec, and current amplitude was measured at the end of hyperpolarizing pulse, while, for those on *I*_K(DR)_, it was depolarized from −50 to +50 mV with a duration of 1 sec, and current amplitude measured at the beginning (i.e., peak *I*_K(DR)_) and end (i.e., late *I*_K(DR)_) of the command pulse was collected. Current amplitudes were measured and compared in control (i.e., CIL was not present) and during cell exposure to varying concentrations (0.1–100 μM) of CIL. The concentration required to suppress 50% of current amplitude (i.e., *I*_h_ and peak or late *I*_K(DR)_) was calculated according to the modified Hill equation:(1)$$Relative\;amplitude=\frac{\left(1-a\right)\times {\left[CIL\right]}^{-n_{H}}}{IC_{50}^{-n_{H}}+{\left[CIL\right]}^{-n_{H}}}+a$$
where IC_50_ or n_H_ is the CIL concentration (i.e., [CIL]) required for a 50% inhibition or the Hill coefficient, respectively, and maximal inhibition (i.e., 1-*a*) was also approximated from this equation. 

To evaluate the impedance (*Z*) of the system, a complex number was defined as:(2)$$Z=Z_{\mathrm{real}}+Z_{\mathrm{imaginary}}=\frac{\mathrm{FFT}\left(V\right)}{\mathrm{FFT}\left(I\right)}$$

The impedance was displayed as the impedance magnitude (IM) against frequency (i.e., IM plot) [29,30]. The resonance frequency (f_R_) was determined as the maximum value of impedance magnitude from the plot.

To evaluate the steady-state activation curve of *I*_h_ or inactivation curve of *I*_K(DR)_ determined by different two-step voltage profiles, the relationships between the normalized amplitude of *I*_h_ or *I*_K(DR)_ and the conditioning potentials with different durations were least-squares fitted to a Boltzmann function of the following form:(3)$$\frac{I}{I_{max}}=\frac{1}{1+exp\left\{\frac{\left[\left(V-V_{1/2}\right)qF\right]}{RT}\right\}}$$
where *I*_max_ represents the maximal amplitude of *I*_h_ or *I*_K(DR)_, *V* is the prepulse potential, *V*_1/2_ the potential at which half-maximal inhibition occurs, *q* the apparent gating charge in the activation or inactivation curve of the current (i.e., elementary charge [*e*] across the membrane electric field between completely closed and open conformations), *F* the Faraday constant, *R* the universal gas constant, *T* the absolute temperature, and RT/F = 25.2 mV.

The inhibitory effect of CIL in *I*_K(DR)_ measured from GH_3_ cells was analyzed by using a state-dependent blocker, which has a larger affinity of binding to the open state of the channel. A minimal kinetic scheme summarizing the interaction between CIL and the K_V_ channel was undertaken as follows:(4)$$C\;\overset{\alpha }{\underset{\beta }{⇄}}\;O\;\overset{k_{+1}*\cdot (\mathrm{CIL})}{\underset{k_{-1}}{⇄}}\;O\cdot \mathrm{CIL}$$
where (CIL) represents the CIL concentration; α or β is the voltage-gated rate constant for the opening or closing of K_V_ channels, respectively; *k*_+1_^*^ or *k*_−1_ the rate constant used for blocking or unblocking by CIL, respectively; (CIL) the CIL concentration; and, C, O, or O·CIL is the resting closed, open, or open-blocked state, respectively.

We evaluated the blocking (i.e., on, *k*_+1_^*^) or unblocking (i.e., off, *k*_−1_) rate constants from the time constants (τ) of the relative blocking (i.e., (*I*_control_ – *I*_CIL_)/*I*_control_) of *I*_K(DR)_ attained during the exposure to different CIL concentrations. Accordingly, the plot of 1/τ versus (CIL) was linear with *k*_+1_^*^ as the slope and *k*_−1_ as the intercept, thus giving the dissociation constant (*K*_D_).

### 4.6. Statistical Analyses

Linear or nonlinear curve-fitting (e.g., sigmoidal or exponential curve) to any given data sets was undertaken using either Microsoft Excel^®^ (Redmond, WA, USA) or OriginPro 2016 (Microcal). To reduce the noise for the IM plot, we employed a five-point averaging algorithm for the smoothing of data points (Gutfreund et al., 1995). Values are provided as means±SEM with sample sizes (n), indicating the number of GH_3_ or H9c2 cells from which the experimental data were collected. The paired or unpaired Student’s *t*-test and one-way analysis of variance (ANOVA) followed by posthoc Fisher’s least-significant difference test for multiple comparisons were analyzed. The data were examined by the nonparametric Kruskal–Wallis test, subject to a possible violation in the normality underlying ANOVA. Differences were considered statistically significant when the *p-*value was below 0.05. 

## 5. Conclusions

The prominent results in the present study were as follows. (a) In GH_3_ cells, CIL could suppress hyperpolarization-elicited *I*_h_ in a concentration-, time-, and voltage-dependent manner. (b) This compound was able to consistently suppress depolarization-elicited *I*_K(DR)_ in combination with the enhanced time course of current inactivation. (c) The IC_50_ required for the CIL-inhibited *I*_K(DR)_ measured at the end of the depolarizing voltage-clamp step was similar to the *K*_D_ value achieved according to the minimal kinetic scheme. (d) As the CIL concentration increased, the steady-state inactivation curve of *I*_K(DR)_ could be shifted toward hyperpolarized potentials with no difference in the gating charge of the current, and the recovery of current inactivation was prolonged. (e) The presence of CIL failed to modify the amplitude or gating of *I*_Na_ during abrupt step depolarization. (f) In heart-derived H9c2 cells, unlike the depressant action of CIL, the addition of neither ivabradine nor zatebradine noticeably altered the inactivation time course of *I*_K(DR)_, while they mildly suppressed current amplitude. The observed effect of CIL on cellular functions thus could not be solely restricted to its suppression of *I*_h_. Another unidentified but important inhibitory effectiveness of CIL in altering the amplitude and gating of *I*_K(DR)_ demonstrated here, at least partly, participated in its regulatory actions, which are likely to occur in vivo. Overall, the concerted ability of CIL to modulate the amplitude and gating of *I*_h_, *I*_K(DR)_, or both was capable of exerting a significant contribution to its modulation in electrical behaviors of the in-vivo endocrine or heart cells.

## Figures and Tables

**Figure 1 ijms-21-02416-f001:**
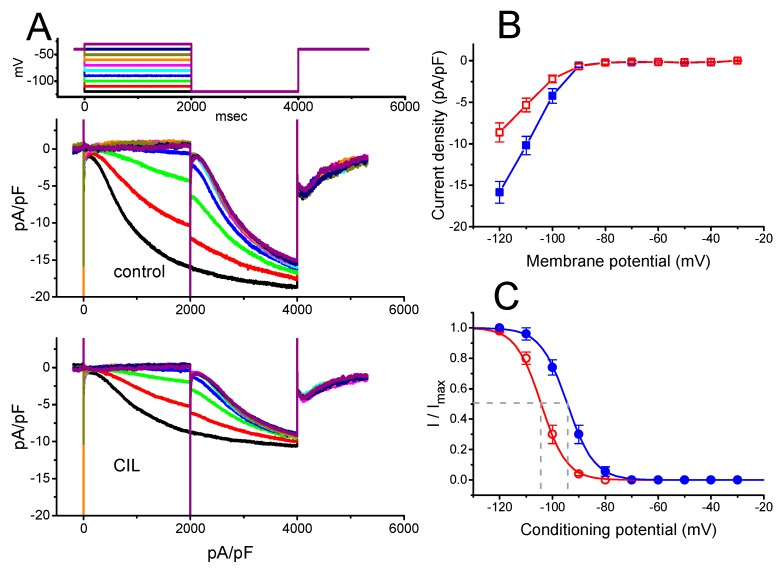
Effect of cilobradine (CIL) on hyperpolarization-activated cation current (*I*_h_) recorded from pituitary GH_3_ cells. We bathed cells in Ca^2+^-free Tyrode’s solution containing 1 μM tetrodotoxin (TTX), and, during the recordings, we filled patch pipette by using a K^+^-containing solution. (**A**) Superimposed *I*_h_ traces obtained in control (upper) and after application of 3 μM CIL (lower). (**B**) Averaged current density versus voltage relationships of *I*_h_ measured in control (■) and after application of 3 μM CIL (□) (mean ± SEM; n = 8 for each data point). The current amplitude was taken at the end of the 2-s step hyperpolarization. (**C**) Steady-state activation curve of *I*_h_ in control (●) and after the CIL (3 μM) addition (○) (mean ± SEM; n = 8 for each data point). The vertical dashed lines are placed at the value of *V*_1/2_ for the inactivation curve. (**D**) Concentration-dependent inhibition of *I*_h_ produced by CIL. Each cell was maintained at −40 mV, and the current amplitude obtained during exposure to different CIL concentrations was then measured at the end of the 2-s step hyperpolarization to −120 mV. The value for IC_50_ (indicated in the vertical dashed line) or Hill coefficient was calculated to be 3.38 μM or 1.1, respectively. Continuous sigmoidal line in (C) or (D), on which the data points were overlaid, represents best fits to the Boltzmann (Equation (3)) or Hill equation (Equation (1)), respectively, as elaborated under Materials and Methods.

**Figure 2 ijms-21-02416-f002:**
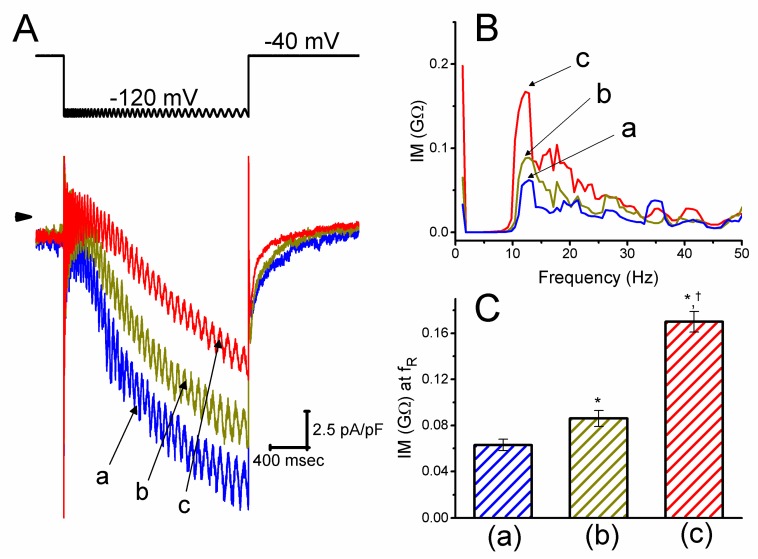
Effect of CIL on *I*_h_ elicited by step hyperpolarization from −40 to −120 mV with changing frequency in GH_3_ cells. (**A**) Representative *I*_h_ densities obtained in the absence (a) and presence of 1 μM CIL (b) or 3 μM CIL (c). The upper part indicates the voltage protocol applied, whereas the arrow is the zero-current level. (**B**) Effect of CIL on the plot of IM (impedance magnitude) taken during the control (a) and after 1 μM CIL (b) or 3 μM CIL (c) application. To decrease noise, each continuous line was smoothened by using a five-point averaging algorithm. (**C**) Summary bar graph showing the effect of CIL on the impedance magnitude (IM) measured at the resonance frequency (f_R_) (mean ± SEM; n = 8 for each bar). The f_R_ value was determined as the peak in the IM plot generated by dividing the Fourier transform of the voltage signal by those of the current signal. ^*^Significantly different from control (*p* < 0.05) and ^ƚ^significantly different from 1 μM CIL alone group (*p* < 0.05).

**Figure 3 ijms-21-02416-f003:**
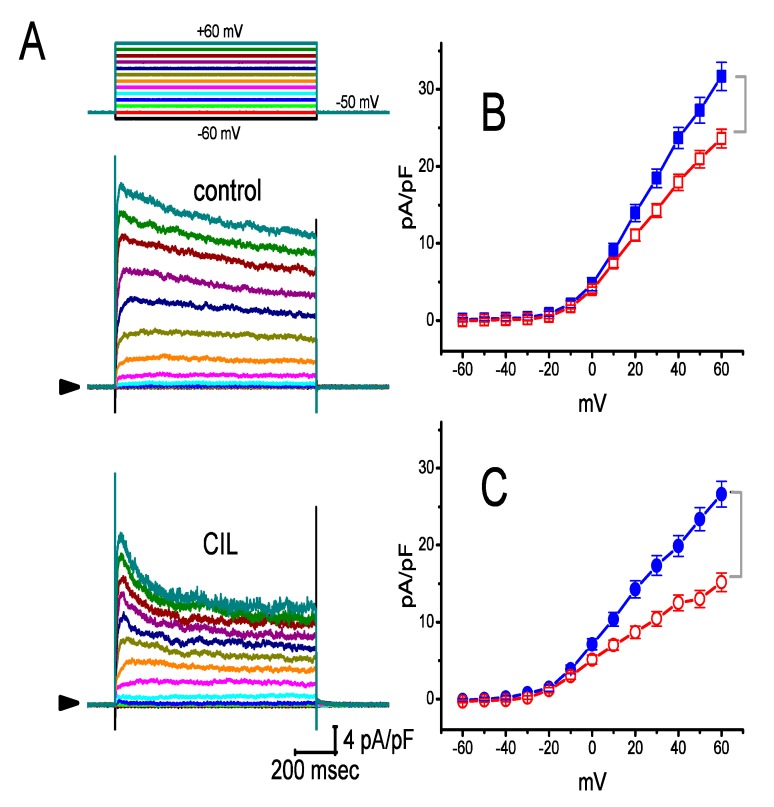
Effect of CIL on delayed-rectifier K^+^ current (*I*_K(DR)_) identified in GH_3_ cells. Cells were bathed in Ca^2+^-free Tyrode’s solution containing 1 μM tetrodotoxin, and the recording pipette was filled with a K^+^-containing solution. (**A**) Representative *I*_K(DR)_ densities obtained in the absence (upper) or presence (lower) of 3 μM CIL. In these experiments, the examined cell was voltage-clamped at −50 mV, and a series of the voltages ranging between −60 and +60 mV in a duration of 1 sec was applied. The voltage-clamp protocol is illustrated in the uppermost part, the arrow shown at each panel indicates the zero-current level, and the calibration mark in the right lower corner applies all current traces. In (**B**) and (**C**), averaged current density versus voltage relationships of *I*_K(DR)_ obtained during the control (filled symbols) and after application of 3 μM CIL (open symbols) were, respectively, measured at the beginning (square symbols, peak *I*_K(DR)_) and end (circle symbols, late *I*_K(DR)_) of each step depolarization (mean ± SEM; n = 8 for each data point). The bracket indicated in the right side of (B) and (C) denotes the difference in magnitude of CIL-mediated decrease of peak and late *I*_K(DR)_ densities taken at the level of +60 mV, respectively.

**Figure 4 ijms-21-02416-f004:**
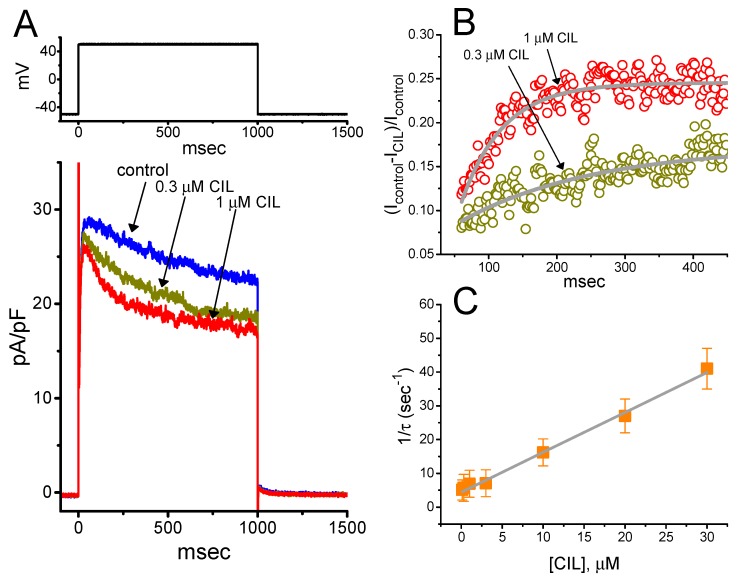
Kinetic analyses of CIL-mediated inhibition of *I*_K(DR)_ in GH_3_ cells. (**A**) Superimposed *I*_K(DR)_ densities in the absence (a) and presence of 0.3 μM CIL (b) or 1 μM CIL (c). The upper part is the voltage protocol used. (**B**) The time course of a relative block of *I*_K(DR)_ during exposure to 0.3 μM CIL or 1 μM CIL was least-squares fitted by a single exponential with the value of 175 or 145 msec (indicated by the smooth line), respectively. The relative block of the current (i.e., (*I*_control_-*I*_CIL_)/*I*_control_) was determined by dividing CIL-sensitive current by the current obtained in control. In (**C**), the reciprocal of time constant (i.e., 1/τ) of relative block versus the CIL concentration was constructed and then plotted. Data points (indicated in filled squares) were approximately fitted by linear regression, indicating that there was a molecularity of one. From the first-order reaction scheme, as elaborated in Materials and Methods (Equation (4)), blocking (*k*_+1_^*^) or unblocking (*k*_−1_) rate constant for CIL-mediated block of *I*_K(DR)_ was calculated to be 1.18 sec^−1^μM^−1^ or 4.45 sec^−1^, respectively. Each data point indicates the mean ± SEM (n = 89–). (**D**) Concentration-dependent inhibition of CIL on *I*_K(DR)_ amplitude measured at the beginning (square symbols) and end (circle symbols) of 1-s depolarizing step to +50 mV from a holding potential of −50 mV (mean ± SEM; n = 89– for each data point). The continuous lines overlaid onto the data points were fitted by the modified Hill equation, as detailed in Materials and Methods (Equation (1)). The IC_50_ value achieved at the beginning or end of 1-s step depolarization (indicated in the vertical dashed line) was estimated to be 9.32 or 3.54 μM, respectively.

**Figure 5 ijms-21-02416-f005:**
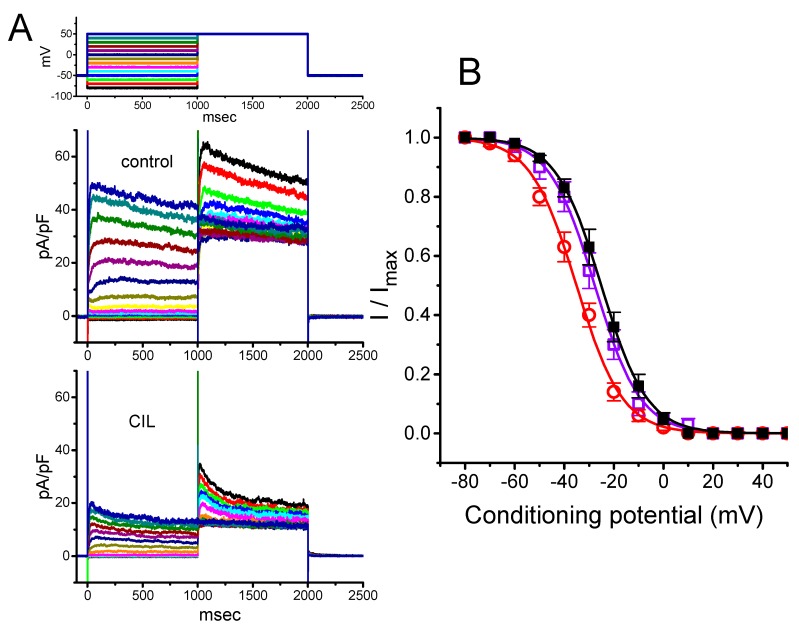
Effect of CIL on the steady-state inactivation curve of *I*_K(DR)_ in GH_3_ cells. In these experiments, a two-step voltage protocol (indicated in the upper part of (A)) was applied to each examined cell. (**A**) Superimposed *I*_K(DR)_ densities obtained in control (upper) and during cell exposure to 3 μM CIL (lower). (**B**) Averaged inactivation curve of *I*_K(DR)_ in the absence (■) and presence of 1 μM CIL (□) or 3 μM CIL (○) (mean ± SEM; n = 7 for each data point). Each sigmoidal curve overlaid was fitted to the modified Boltzmann equation, as elaborated in Materials and Methods (Equation (3)).

**Figure 6 ijms-21-02416-f006:**
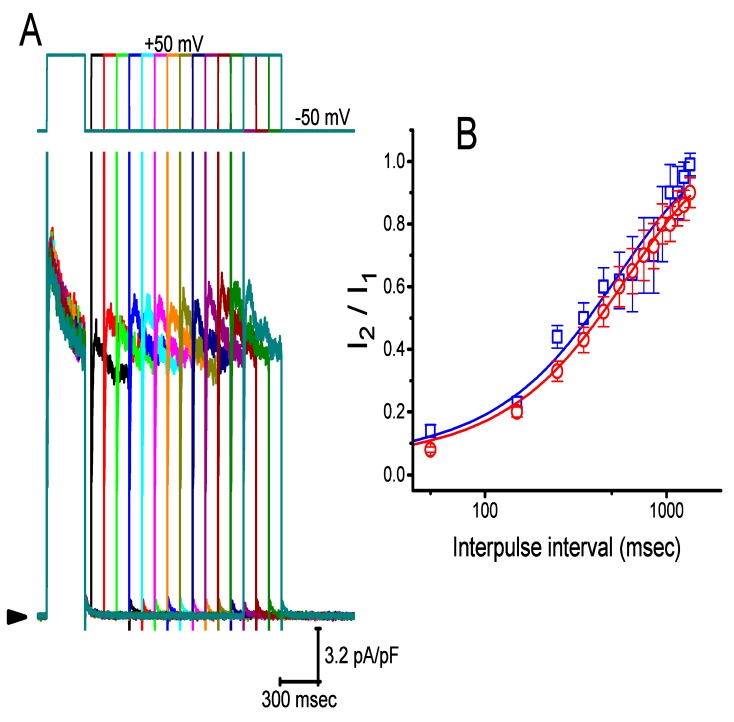
Effect of CIL on recovery from the inactivation of *I*_K(DR)_ after the cell was depolarized from −50 to +50 mV. (**A**) Superimposed *I*_K(DR)_ densities evoked by varying inter-pulse intervals (as indicated in the upper part) in the presence of 3 μM CIL. Arrowhead indicates the zero current level. (**B**) Time course of recovery of *I*_K(DR)_ inactivation in the presence of 1 μM CIL (□) and 3 μM CIL (○) measured from GH_3_ cells (mean ± SEM; n = 8 for each point). I_1_ or I_2_ indicates the peak amplitude of first or second *I*_K(DR)._

**Figure 7 ijms-21-02416-f007:**
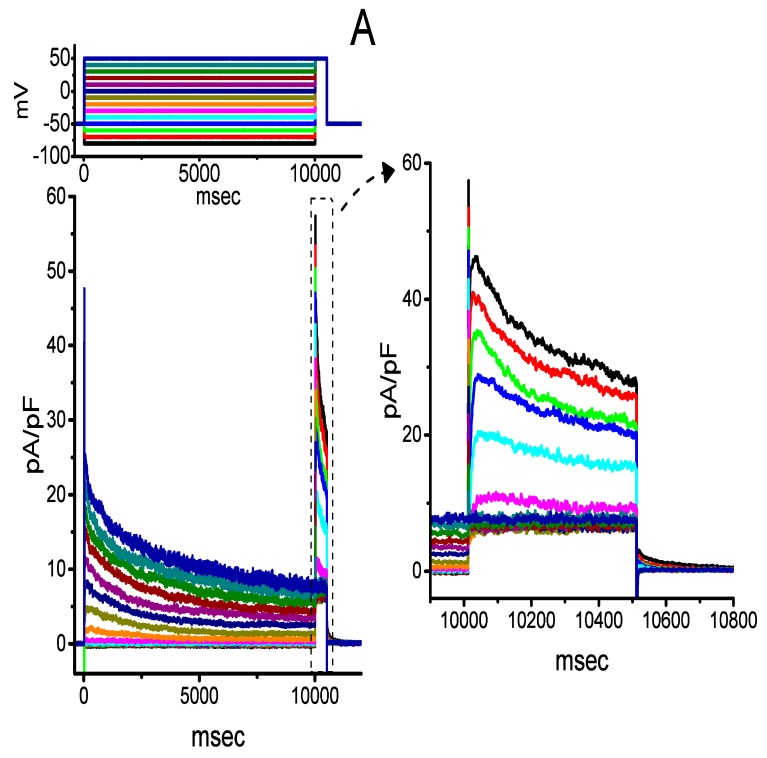
Effect of CIL on the steady-state inactivation curve of *I*_K(DR)_ elicited by 10-s conditioning pulses observed in GH_3_ cells. (**A**) Superimposed *I*_K(DR)_ densities obtained in the presence of 3 μM CIL. The conditioning voltage pulses with a duration of 10 sec to various membrane potentials were delivered, and the voltage protocol applied is indicated in the upper part. The current-density traces on the right side indicate an expanded record from the left side (indicated in dashed box). (**B**) Steady-state inactivation curve of *I*_K(DR)_ obtained in the presence of 1 μM CIL (□) and 3 μM CIL (○) (mean ± SEM; n = 7 for each data point). The vertical dashed line is placed at the *V*_1/2_ value of the inactivation curve of the current.

**Figure 8 ijms-21-02416-f008:**
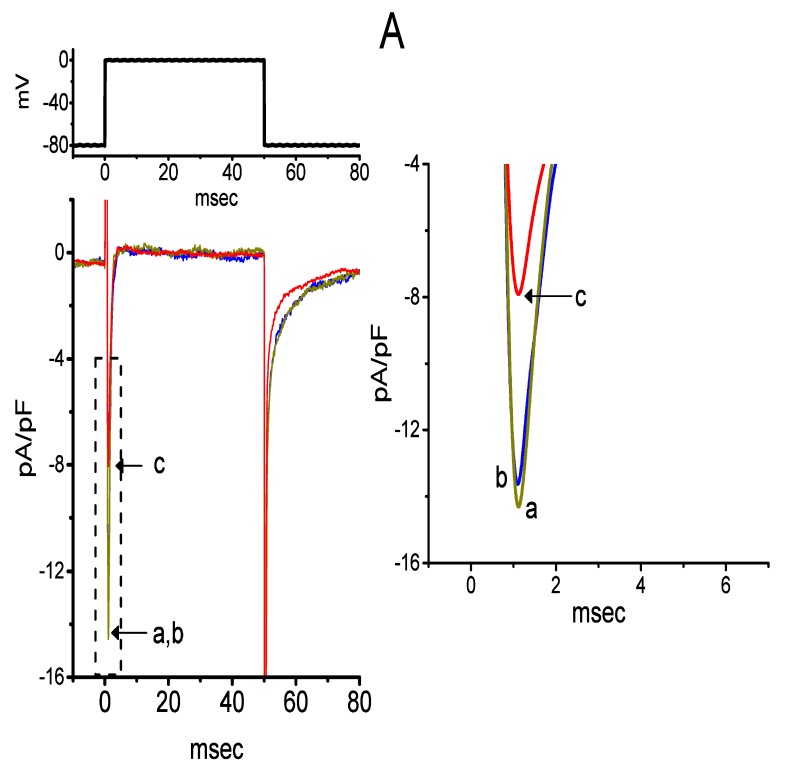
Effect of CIL on voltage-gated Na^+^ current (*I*_Na_) in GH_3_ cells. In this set of whole-cell current recordings, we suspended cells in Ca^2+^-free Tyrode’s solution containing 10 mM tetraethylammonium chloride TEA, and the recording pipette was filled with Cs^+^-containing solution. (**A**) Superimposed *I*_Na_ densities in response to brief membrane depolarization from −80 to 0 mV. The current-density trace labeled a was control, that labeled b or c was recorded in the presence of 3 μM CIL, or 3 μM CIL plus 3 μM A-803467, respectively. The upper part indicates the voltage protocol applied. The density traces shown in the right side depict an expanded record at the left side (indicated in dashed box). (**B**) Summary bar graph of the effect of CIL, CIL plus A-803467, and CIL plus tefluthrin (Tef) on the peak density of *I*_Na_ during brief step depolarization from −80 to 0 mV (mean ± SEM; n = 8–9 for each bar). CIL: 3 μM CIL; A-803467: 3 μM A-803467; Tef: 3 μM tefluthrin. ^*^Significantly different from control (*p* < 0.05).

**Figure 9 ijms-21-02416-f009:**
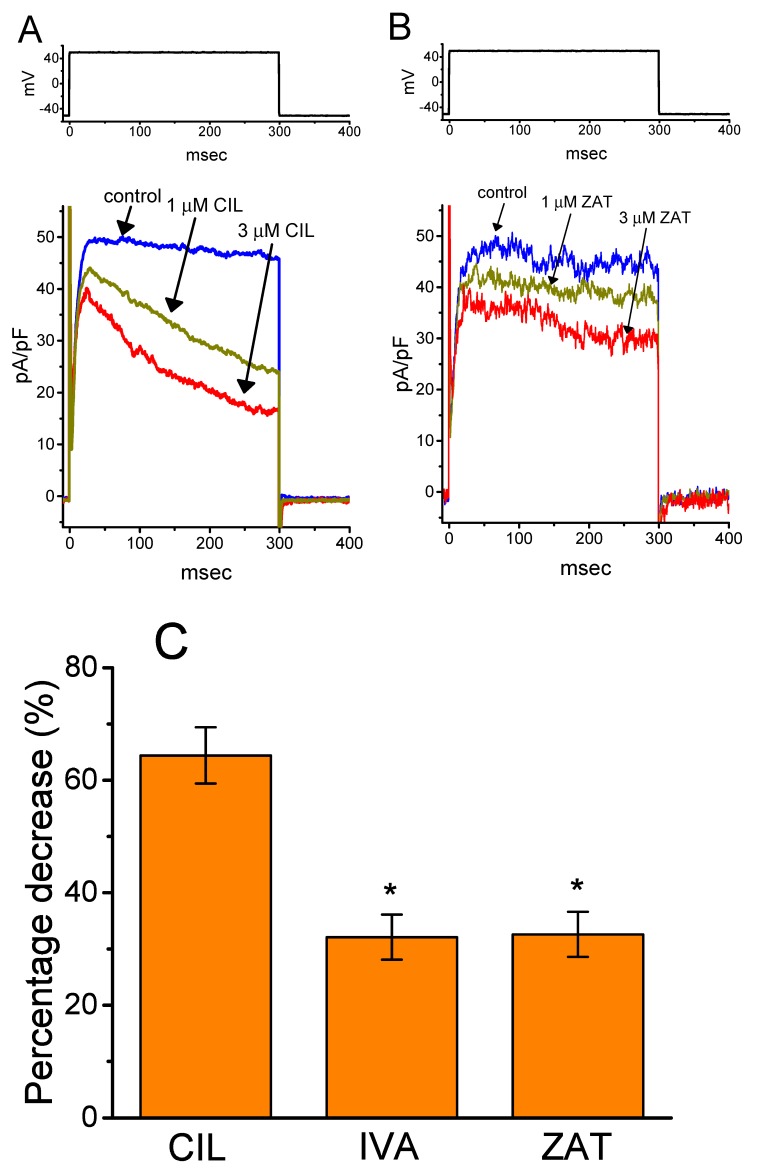
Effect of CIL, ivabradine (IVA), or zatebradine (ZAT) on depolarization-elicited *I*_K(DR)_ densities recorded from H9c2 cardiac cells. Cells were bathed in Ca^2+^-free Tyrode’s solution containing 1 μM tetrodotoxin, and we backfilled the recording pipette with a K^+^-containing solution. In (**A**)¸ *I*_K(DR)_ density elicited by 300-msec step depolarization from −50 to +50 mV was obtained in control and after addition of 1 μM CIL or 3 μM CIL, while in (**B**), that was taken in the absence and presence of 1 μM ZAT or 3 μM ZAT. The upper part in (A) or (B) is the voltage protocol used. (**C**) Summary bar graph of inhibitory effects of CIL (3 μM), IVA (3 μM), or ZAT (3 μM) on percentage decrease of *I*_K(DR)_ in H9c2 cells (mean ± SEM; n = 8). The current density was measured at the end of the 300-msec depolarizing pulse from −50 to +50 mV. ^*^Significantly different from CIL (3 μM) group (*p* < 0.05).

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
