# Peer review of "Inhibitory Effective Perturbations of Cilobradine (DK-AH269), A Blocker of HCN Channels, on the Amplitude and Gating of Both Hyperpolarization-Activated Cation and Delayed-Rectifier Potassium Currents"

_ijms, 2020, doi:10.3390/ijms21072416_

Round 1

Reviewer 1 Report

This is interesting and important and relevant research that specificity and potency of CIL, DK-AH269, which is promising therapeutic compound.

  1. Selection of H9c2 cell line is well justified. But, why were GH3 pituitary tumor cells, but not other excitable cells selected?
  2. Detailed protocols and justification to distinguish Ih and IK(DR) from other currents should be in method section. Method section should also contain all used bath and pipette solutions.
  3. Have double exponential equation tried for fitting. Boltzmann question is not always ideal.
  4. Could you record Ih from H9c2 cells?
  5. Data in Fig 5 and 6 are over-interpreted. There is no any statistically significant effect of CIL on inactivation and the steady-state inactivation curve of IK(DR).
  6. Generally, set of experiments should be more justified. For example, why was “the steady-state inactivation curve of IK(DR) evoked by 10-sec conditioning pulses” measured?
  7. Do GH3 cells contain NaV1.8 channels?  

All-in-all, the manuscript should contain more information as specified above.

Author Response

Reviewer #1 (ijms-744347)

        We are grateful for the reviewer’s comments on our work.

        The reply in the revised manuscript follows the sequence of the specific comments raised by the reviewer. We also upload the revised manuscript and the changed text was indicated in the red color.

Reviewer 2 Report

This paper describes the inhibitory effect of cilobradine (CIL) on HCN channel currents (Ih) as well as on delayed-rectifier potassium currents (IK(DR)) in pituitary tumor (GH3) cells and heart-derived H9c2 cells. In in vitro electrophysiological studies focused on detailed whole-cell current analyses. Specifically, current response patterns, impedance amplitude (as a function of frequency), kinetics, steady-state inactivation (1 s and 10 s) were analyzed. CIL suppressed the whole-cell currents in a dose-dependent matter and the steady-state activation slightly shifted to more negative potentials. Furthermore, the inhibitory effects of other bradycardiac agents such as zatebradine and ivabradine on whole-cell delayed-rectifier potassium currents in H9c2 cells were also investigated and compared to the effect of CIL, whereby CIL had the strongest effect. Incidentally, no effect on sodium channels could be detected. The authors concluded that CIL is able to change the amplitude and gating of IK(DR) linked with functional activities of electrical excitable GH3 cells. Generally, this manuscript is well written and understandable but its content and form can be improved. Although the abstract contains information on the research background, it does not clearly lead to a primary question, methodic and main results as well as main conclusions. In the introduction, the relevance of the topic and derivation of the scientific issues were not clearly explained. The description of the method contains information gaps (e.g. normalization of whole cell current amplitudes, cell membrane capacitance). On the other hand, the result section is well structured. The results are understandable and transparently delineated (although some minor issues can be clearly improved). The discussion start with a short summary of the main results, which is good. Data are well compared with the current literature whereas the limitations of this study were not discussed.

Specific comments

Abstract

What was the rationale to investigate two different cell types and to investigate two different kind of currents? It should be at least mentioned that the (conventional) whole-cell patch-clamp technique was used.

Method

For analyzing the effect of drugs or compounds like CIL in different cell types, the cell membrane capacitance should be considered. Accordingly, current densities (pA/pF) should be available and can be used to compare electrophysiological characteristics between different cell types or different conditions. Before, the mean membrane capacitance (pF) of both cell types should be given in the method section. How was the mean access resistance regarding the quality of the recordings? It is also suggested to give the junction potential. Was it calculated and/or measured? Were leak currents substracted?

Results

As mentioned before, it is suggested to give normalized currrents

Fig. 1: For the voltage stimulation protocol, a more detailed illustration may help to better follow the current response patterns (compare Fig. 4A, in which it is better shown). How long were the voltage pulses (from -120 to -30 mV)? In addition, it is suggested to give the concentration of CIL in the diagram (although it is mentioned in the legend). Panel B: Which time window of the traces shown in panel A were analyzed? The information that it is related to the end of 2-sec step hyperpolarization may be too imprecise. Alternatively, drawing the time frame or highlight the data from panel A, which are shown in the current voltage relation may help in this matter. Furthermore, the traces for control and CIL should be labelled in the diagram. The same applies to panel C. Which data from which diagram in panel A were analyzed? Panel D: It is suggested to show a horizontal reference line at 0.5 (relative amplitude) so that the corresponding concentration of CIL can be roughly read showing a vertical reference line.

Line 280-281: The comparison of data from the literature (Stieber et al., 2006) should be shifted to the discussion.

Fig. 3 (minor): Panel A: How long was the voltage stimulation? Alternatively, a voltage protocol and a current response pattern can be shown as illustrated in Fig. 4A containing a X- and Y axis. In addition, the concentration of CIL should be drawn in the diagram. Panel B and C: See the suggestions of panel B and C in Fig. 1. Is there any explanation of the remaining currents in the presence of 3 µM CIL.

Fig. 4A (current response pattern): It would be better to label the traces with control, 0.3 µM and 1 µM CIL instead of letters. Panel b: Same suggestion (label with 0.3 µM and 1 µM instead of 1 and 2).

Panel D: Reference lines (at 0.5) should be drawn into the diagram (one horizontal and two vertical lines)

Fig. 5: Panel A: The current response patterns of control and 1 µM CIL are missing. Panel B: The traces should be labeled with the CIL concentrations.

Fig. 6 (minor): The cell type should be given in the legend.

Fig. 7: Panel B: From which window frame in panel A were the data in panel B are coming from? Reference lines should be shown (see above). Panel B: The traces should be labeled with the CIL concentrations.

Fig. 8 (minor): Why was TTX not used as a (general) sodium channel blocker?

Fig. 9: Panel A and B: Instead of a, b, c, control, 1 µM CIL and 3 µM CIL should be drawn in the diagram. The measurements (traces) with ZAT are missing.

Discussion

Generally, a discussion figure by figure would be helpful. Alternatively, citing the figure which is discussed in a paragraph may help in this matter.

Line 512 – 515: This speculation would be better supported if normalized currents would be considered. Irrespective of that, a discussion of the limitation of the study is missing.

Overall, the results predominantly contain patch-clamp data from GH3 cells (8 from 9 figures). For comparison purposes, a bar chart summarizing the current densities (normalized Ih and IK amplitudes) of both cell types may help in this matter. Is there any rationale to omit similar measurements of H9c2 cells? At least, mRNA data (PCR) and/or protein data (e.g. immunohistochemistry) of the HCN channel(s) may help to support the conclusion that these channels are clearly expressed in the used cell lines.

Minor

Page 5: It is recommended to give numbers for the equations.

Author Response

Reviewer #2 (ijms-744347)

        We are grateful for the reviewer’s comments on our work.

        The reply in the revised manuscript follows the sequence of the specific comments raised by the reviewer. We also upload the revised manuscript and the changed text was indicated in the red color.

Round 2

Reviewer 2 Report

The manuscript is considerable improved in particular regarding the formal issues with the figures and the patch-clamp method.

  • Additional data with respect to cell membrane capacitances and series resistances in used cells were correctly included in the manuscript.
  • The missing information in the method section is now included. Current densities are now given so that a comparison between different cell types are possible. The information of the maximal leak current amplitudes which were subtracted would be interesting to know. 
  • In all figures, current densities (pA/pF) are now given which is crucial for comparison purposes.
  • For the voltage stimulation protocols in the corresponding figures, more detailed illustrations are now given helping to better follow the current response patterns.
  • Control currents in Fig. 5 are added.
  • Labeling of the figures are improved, in particular Fig. 9
  • Figures are cited in the discussion.
  • The protein or mRNA expression to validate these HCN channels are not given. Since the authors are working with cell lines and measure whole-cell currents, why would they assume that these channels are clearly expressed in the used cell lines and that only these channels are expressed in the whole-cell currents. Are there similar studies in the literature using these cell types and current densities are comparable? See Schultz, J.H.; Volk, T.; Ehmke, H. Circ Res 2001.